# FUNCTIONAL SEGREGATION OF INPUTS IN ARTIFICIAL NEURAL NETWORKS FOR VISION

## ABSTRACT

One of the main organizational principles of artificial and biological intelligence systems is their reliance on signed inputs: positive and negative weights in artificial networks, and excitatory and inhibitory synapses in the brain. However, little is known about the role of inhibitory activity in high-level visual cortex such as inferotemporal cortex, or how artificial neural networks (ANNs) trained for object recognition segregate their learned representations into positive and negative weights. Here, we dissected high-level visual mechanisms in ANNs trained with ImageNet. We investigated how learned representations of ANN classification units depended on their positive or negative inputs using ablation experiments and feature visualization. We found that unit representations changed more when ablating positive- vs. negative inputs. Object-related features were abolished when ablating positive inputs, while still preserving background textures. This effect was more pronounced in adversarially trained robust networks. This segregation persisted in networks trained with unsupervised learning, but was not present in a ResNet18 trained with Tanh instead of ReLU. We found a consistent functional segregation when we trained models to replicate the activity of neurons in monkey visual cortex, across the ventral stream (V1, V4, and IT). Feature visualization of the neuron models produced images containing local features preferred by actual neurons. Analogous to units trained for classification, the learned representations of units trained to simulate neurons changed more upon ablating positive than negative inputs. We conclude that ANNs for classification segregate object or foreground information into the positive weights, with background or contextual information into the negative weights, in their last layer before softmax. These results hint at the relevance of signal rectification and inhibition into shaping feature selectivity in the primate ventral stream, a hypothesis we are testing in vivo.

## 1 INTRODUCTION

Artificial and biological intelligence systems rely on units that influence each other via signed mechanisms: hidden units interact via positive and negative weights in artificial networks, and neurons interact via excitatory and inhibitory synapses in the brain. Cortical neurons divide into types defined by their genetic, anatomic and functional properties (Zeng, 2022). Excitatory neurons are thought to compute the main features, and inhibitory neurons are thought to provide contextual information to excitatory neurons, gate and route the information in cortical circuits. Object classification in humans relies on the occipitotemporal visual system, the "ventral stream". Neurons in the ventral stream are selective to more complex visual features along the hierarchy, similar to how features increase in complexity along the depth of convolutional neural networks (CNNs) trained for object classification. In the early visual system, including the retina, lateral geniculate nucleus, and primary visual cortex (V1), inhibitory neurons provide lateral inhibition, which defines the center-surround receptive field organization. Lateral inhibition spatially sharpens receptive fields and enhances feature selectivity by suppressing redundant information in the surround of the excitatory receptive field center. This is clear when the excitatory receptive field features are spots or sine-wave gratings. However, the role of inhibition in the highest levels of the hierarchy in the primate ventral stream remains unkown, particularly in V4, posterior, central, and anterior inferotemporal cortex (pIT, cIT, aIT).

ANNs compute with positive and negative weights, analogous to excitation and inhibition in the brain. However, it is not known how visual information is parsed across positive and negative weights. Particularly, for object classification networks, it is not clear how the selectivity to different object categories emerges in their output layers, where each unit corresponds to one object category. One hypothesis is that information is largely segregated across absolute weight strengths (Li et al., 2023). Because classification CNNs are good models of the ventral stream, we hypothesize that CNN units might also segregate different kinds of visual information into their positive and negative input weights.

To test the hypothesis of functional segregation across positive and negative weights in classification networks, we performed ablation experiments and feature visualizations of output units of different ImageNet-trained CNNs. By performing ablations of varying magnitude, we studied how visual information was organized across and within excitatory and inhibitory weights, and if and how different ranges of weight strengths corresponded to different parts of objects and backgrounds. We found that both positive and negative ablations resulted in activity changes, but only the ablation of positive weights significantly changed the preferred images obtained from feature visualizations. The ablation of negative inputs resulted in images with contextual variations, i.e., changes in colors of the object in the foreground or in the background. For example, negative input ablation resulted in images with white backgrounds for a robust ResNet50. Gradual removal of positive inputs produced gradual deformations of the object, but not in a parts-based fashion. Surprisingly, such representational changes upon input ablations were more pronounced in robust networks, which are trained to resist noise perturbations to images (Szegedy et al., 2014; Salman et al., 2020; Elsayed et al., 2018). Total positive input ablation led to the removal of objects while preserving background features, this was also quantified by an object detection network, YOLOv7 (Wang et al., 2022). To test if this functional segregation generalized to other objective functions such as prediction, we used the same networks as models that predicted the images responses of neurons in the ventral stream. The models were simply a re-weighting of the same inputs to the final fully-connected layers of ImageNet CNNs obtained via linear regression between the neuron responses and the neural network penultimate layer activations. Thus, if neurons had the same object bias as CNNs, they will show a similar functional segregation to CNNs. And if neurons did not have such object bias, but still showed a functional segregation, it would reveal another form of functional segregation. To identify their preferred features of the biological neurons that were modeled, we used a (model-free) closed-loop image synthesis approach that bypassed the CNN-fitting stage (Ponce et al., 2019). Feature visualization of the neuron-fitted CNN models (*neuron-model units*) produced images containing features that were also exciting to the the neuron and qualitatively similar to the biological model-free features. Neurons responded more to the visualized preferred images of the model than to the natural images used for model training. Consistent with the classification units, neuron-model units were also more robust to negative than positive input ablation. Yet, unlike the classification units, the preferred images of neuron model units did not seem to contain objects. In sum, our work reveals that for object classification and neuron model units, respectively, the foregrounds or preferred visual features are represented in the positive input weights, while the backgrounds or contextual features are represented in the negative input weights.

## 2 RELATED WORK

**Mechanistic interpretability of computer and biological vision**   There has been progress in mechanistic interpretability in ANNs from work using perspectives from circuit dissection, like those in neuroscience (Olah et al., 2020). This area of explainable artificial intelligence explains model behavior by leveraging smaller network subgraphs to identify relevant features, how they arise from input weights, and how they can be used to build new features hierarchically. It has revealed motifs of positive and negative connections between related features that resemble the organization of the early visual system. Related work has focused on characterizing the object-shape- and texture-biases in feature visualizations by choosing sparse sets of weights to reconstruct individual images (Li et al., 2023). Here we focus on the division between positive and negative inputs across the whole range of weight strengths, which was not covered in that study.

**Feature visualization by closed-loop optimization**   To understand the information learned by neural networks, it serves to analyze their learned representations. For vision networks, both bio-

logical and artificial, the longest-standing approach is to generate images that strongly activate individual units, either by hand (Hubel & Wiesel, 1959) or more recently, using discriminative and/or generative networks. *In silico*, where gradients are available, features can be obtained by performing gradient-ascent from the target unit to the image pixels (Erhan et al., 2009; Nguyen et al., 2016a;b; Olah et al., 2017). Because gradients are not available *in vivo*, gradient-free algorithms have been developed to optimize images preferred by biological neurons in real-time (Ponce et al., 2019; Xiao & Kreiman, 2020; Wang & Ponce, 2022). These gradient-free algorithms rely on black-box optimization of an input to a generative adversarial network (GAN). This constraints the image search space to the natural image priors learned by the GAN, avoiding high-frequency noise which can also be highly activating to the target unit but does not seem to relate well to natural images (Nguyen et al., 2016a). Other approaches rely on first training a neural network to predict neuronal responses to images, and then performing gradient-ascent on the network (Bashivan et al., 2019; Walker et al., 2019). Those methods have been largely used with gray-scale images. Here, we deal with color images using the gradient-free approach in both our investigations in CNNs and our experimental recordings of non-human primates.

**Robustness** Neural networks are susceptible to adversarial attacks, where noise that is nearly imperceptible by humans can be added to natural images, changing output classification (Szegedy et al., 2014; Salman et al., 2020; Elsayed et al., 2018). A proposed solution to the adversarial attacks is robust training, which introduces noise (or another attacks) into the training phase of the network with the aim to build resistance against that particular attack. In theory, robust networks should function more like the primate brain, which shows limited vulnerability to such attacks. Here, we study how trained robustness relates to image representations after weight ablations.

## 3 METHODS

An extended methods section is in the Appendix A.1.

**Networks** We performed our ablation studies in CNNs pretrained on the ImageNet dataset: AlexNet (Krizhevsky et al., 2012), VGG16 (Simonyan & Zisserman, 2015), ResNet50 (He et al., 2015), and robust ResNet50 ($L_\infty \in \{0.5, 1, 2, 4, 8\}$, Salman et al. (2020)). To reduce computing time, we used the *imagenette* dataset (Fas, 2024) and the *ImageNet* macaque category. For all networks, we visualized the representations of the units in the fully-connected output layer (pre-softmax) matching those classes under different ablation conditions.

**Ablation** We ablated weights that were either (1) only positive or (2) only negative. We used a cumulative approach: we first sorted the positive (or negative) weights by their (absolute) decreasing value. Then, we defined a fraction of the total positive or total negative weights to ablate $\alpha$ (*ablation strength*), identifying the top $k$ weights such that $\frac{\sum_{i=1}^{k} w_i}{\sum_i w_i} \leq \alpha$, and set them to zero. We covered the range of ablations from 0 to 1.

**Feature visualization** For each ablation condition, we performed feature visualization by optimizing a GAN latent code to create an activity-maximizing image. We used this closed-loop, zeroth-order-search approach to allow comparison with our neuronal experiments, where gradient ascent would not be possible. To increase the span of the stimulus space, we used two GANs: AlexNet fc6 DeePSiM (Dosovitskiy & Brox, 2016) which can render textures and objects, and BigGAN (Brock et al., 2019) that can render photo-realistic images with objects. For optimization, we used a variant of *covariance matrix adaptation evolutionary strategy* or CMAES (Wang & Ponce, 2022; Loshchilov, 2015). We optimized ten images per GAN, resulting in 20 feature visualizations per output unit and ablation condition. Diverse visualizations better capture the multifaceted high-level representations in CNNs (Nguyen et al., 2016b). For our examples, we show the best of the 20 visualizations, but used all for quantitative analyses. For visualizations of neural networks predicting biological neuron responses, due to experimental time restrictions, we used five visualizations per ablation condition, via DeePSim only. Our experiments are performed in a PC with Nvidia 4090 GPU, and each visualization takes about 3 mins.

**Feature analysis**   We computed image similarity using an ensemble of CNNs, including AlexNet, ResNet50, and ResNet50 with robustness in $L_\infty \in \{0.5, 1, 2, 4, 8\}$, inspired by (Feather et al., 2023) And confirmed the results with LPIPS (Zhang et al., 2018) in the appendix. We computed their activations and defined similarity as the average pairwise cosine similarity between control activity vs input-ablated activity. We averaged the results over all networks. We computed *objectness* as the maximum bounding box score provided by YOLOv7 (Wang et al., 2022).

**Visual cortex electrophysiology**   We collected data from two animals (monkey C and monkey D), each implanted chronically with multielectrode arrays of 32 or 16 channels (monkey C, N = 96 electrodes, monkey D, 64), in areas V1, V4 and posterior inferotemporal cortex (PIT). Some electrodes captured the activity of single units, but most showed multi-unit activity (reflecting the pooled activity of microclusters of neurons). The animals performed a simple fixation task, which required them to keep their eyes on a 0.25-diameter spot at the center of the screen, within a square fixation window measuring 0.7–1° per side. Images were presented for 100 milliseconds ON, 150-ms off, 4-5 images per trial, after which the animal received water or juice.

**Image dataset**   We collected a reference image dataset to activate neurons in the monkey along the hierarchy of V1, V4, and PIT. Because neurons vary in their preferred features, we constructed a dataset spanning the image space as represented by the neural embedding of ImageNet-trained AlexNet. The embedding is the output of the last layer before softmax of AlexNet, a vector space of 1000-dimensions. The images from this dataset also spanned uniformly the 1000-dimensional output space of a semi-supervised trained network, trained on a billion images, ResNet50SS (Yalniz et al., 2019). To define this embedding space, we performed PCA on the output activations from AlexNet to the 50k ImageNet validation images, we kept the top 300 components (accounting for about 95% of total explained variance). Then we partitioned the space into a defined number of clusters $k$, according to the desired dataset size, using batched k-means to reduce computational burden. After finding the $k$ cluster centers, we could feed arbitrary images to the network, map them to the PCA space, and then pick the nearest neighbors to the cluster centers from the desired image space. In addition to the ImageNet validation set, we added other common neuroscience datasets (Brady et al., 2008; Kar et al., 2019; Allen et al., 2022; Hung et al., 2005) to form our image space. We selected $k = 160$ images, as a set that was diverse but small enough to be used in every experimental session. We called this image dataset *diverseSet* .

**Models fit on neuronal activity**   We recorded responses of many neurons in the ventral stream to diverseSet. We performed partial least-squares linear regression (80/20 train/test split) between the neuron responses to images and the activations of the penultimate layer of AlexNet. We selected one neuron or microcluster per experimental session, and performed the ablation and feature visualizations *in silico*. Whenever possible, we also performed the feature visualization of the modeled neuron *in vivo* using a gradient-free approach (Ponce et al., 2019), within the same experimental session. To test whether features learned by the model were relevant to the biological neuron, we recorded the neuronal responses to the preferred images of the model.

## 4 RESULTS

### 4.1 NETWORKS TRAINED ON IMAGENET ALLOCATED OBJECT INFORMATION INTO POSITIVE WEIGHTS

**Hypothesis** The visual system often organizes excitation and inhibition into the center and surround of receptive fields, where the surround is inhibitory and provides contextual information. Thus, we hypothesized that the output units of neural networks for object recognition and classification would also segregate object information to the positive weights and background/contextual information to the negative weights. We tested this hypothesis with ablation experiments and feature visualization in CNNs pretrained in ImageNet.

While units differ in their input weight distribution across networks, all have a total ratio of positive and negative weights close to one. Thus, weights are balanced across polarities (Table 2), supporting the notion that both weight polarities contain relevant information for object recognition and classification.

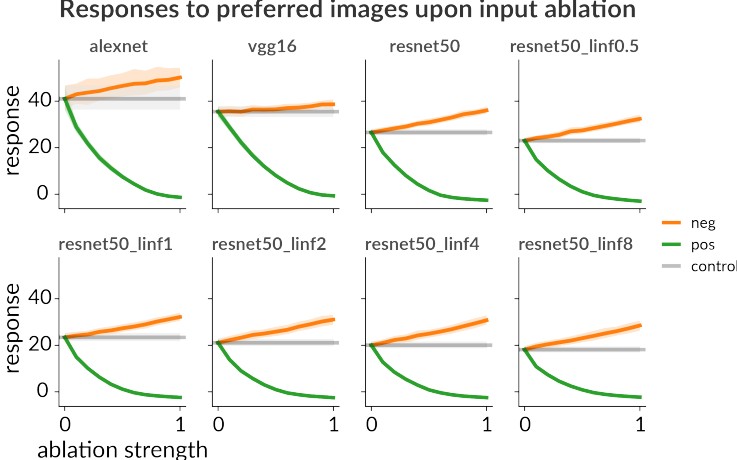

Figure 1: Mean activation scores of units used in ablation experiments. For all networks, units scores come from the last fully-connected layer, with 1000 units, before the softmax. The units correspond to the 10 imagenette categories ([0, 217, 482, 491, 497, 566, 569, 571, 574, 701]) plus the macaque category (373). Error bars are 95% confidence intervals over units (categories tested), where each unit response is the mean of its 20 visualizations. *Control* refers to the feature visualizations in the intact networks for the same units, we extended it as a horizontal line to ease visual comparisons to the different ablation strengths.

To test the visual information encoded into the positive and negative weights, we performed feature visualization of the units under different ablation strengths of only positive or only negative weights. Ablation of positive input weights decreased the maximum achievable activation of the feature visualization, while ablation of negative input weights slightly increased it (Fig. 1). This indicated a functional difference in the contributions of positive and negative input weights to the learned features.

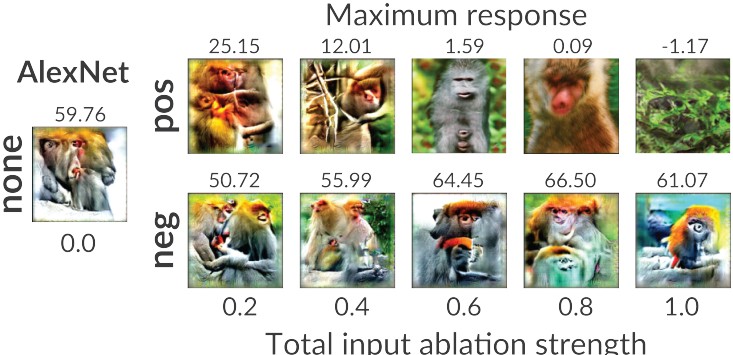

Figure 2: Preferred feature changes for different ablation strengths of input weights to the macaque 373 output unit of AlexNet (last fc layer of 1000 units before softmax). Images are the most activating images out of the 20 visualizations per ablation strength. Ablation strengths are below each image, and activation scores are above.

To identify the functional contributions to the learned representations, we examined the images (see macaque unit example Fig. 2). Visual inspection revealed images changed more with ablation of positive vs negative inputs (Fig. 3). We quantified the changes in representation elicited by ablations as the mean pairwise cosine similarity between images from intact units and images from input-weight-ablated units. Indeed, ablating positive input weights produced representations that differed more from the original representations, while ablating negative input weights resulted in similar representations (Fig. 4). We further verified the changes induced by ablations on the representations of units corresponding to ImageNet classes by doing experiments using a 10x larger dataset made

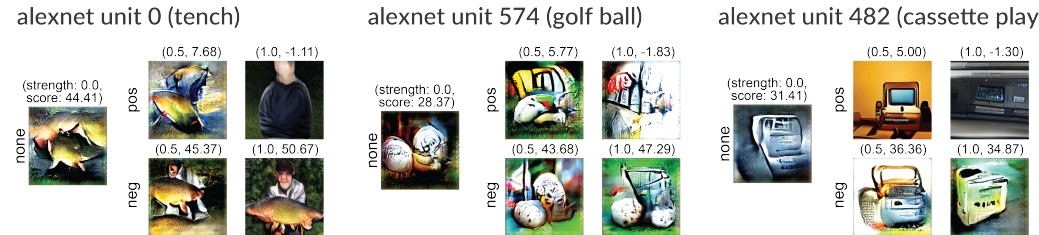

Figure 3: Changes in preferred features to different ablations of example AlexNet output units. Units from the last fully-connected layer of AlexNet, before softmax: 0 tench, 574 golf ball, and 482 cassette player. Each image is the most activating image out of 20 feature visualizations, above image is (ablation strength, activation score) for strengths 0, 0.5 and 1. Top row shows positive ablations, bottom row shows negative ablations. Notice the large image changes for positive ablations.

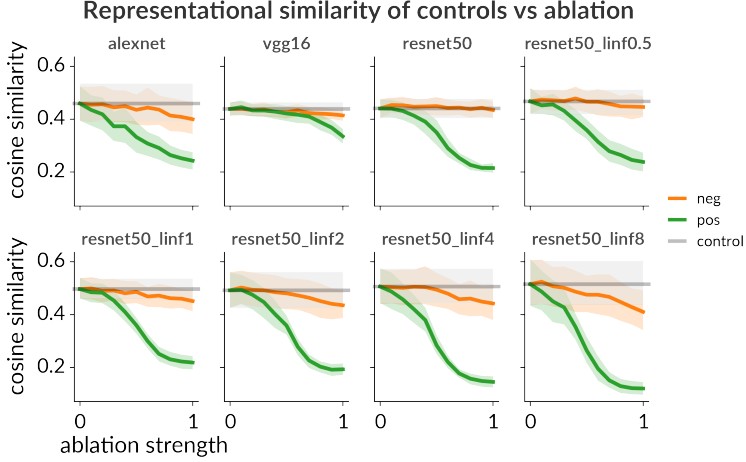

Figure 4: Representational similarity of intact vs input-ablated units across recognition networks tested, measured by the pairwise cosine similarity of control vs ablation images over an ensemble of networks. Error bars are 95% confidence intervals over units, each unit is the mean of its 20 visualizations. The units correspond to the 10 imagenette categories ([0, 217, 482, 491, 497, 566, 569, 571, 574, 701]) plus the macaque category (373).

out of 100 ImageNet classes Fig. 14, also reproduced using a different representational similarity metric, LPIPS (Zhang et al., 2018) Fig. 15. Thus, ablation of positive but not negative weights significantly changes the output representations of ImageNet-trained CNNs.

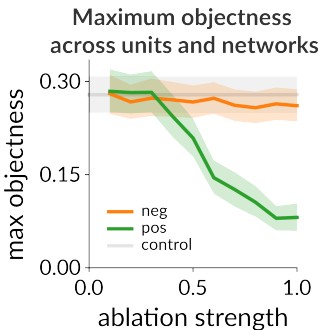

Figure 5: Objectness scores across units per ablation condition. As in previous figures, we tested 11 units from the 1000-unit fully-connected output layer (pre-softmax) of: AlexNet, VGG16, ResNet50, and robust ResNet50 ($L_\infty \in \{0.5, 1, 2, 4, 8\}$). For each network, we averaged over the objectness scores of 20 visualizations per unit and all units. The plot shows the mean over previously described network averages. Error bars are 95% confidence interval over network averages.

Because we observed that complete ablation of positive weights often led to representations lacking the object related to the category, we hypothesized the positive weights encode the object information. To quantify to what extent objects disappear from the preferred images under ablations, we used an object-detection network, YOLOv7 (Wang et al., 2022). Because the CNN units where

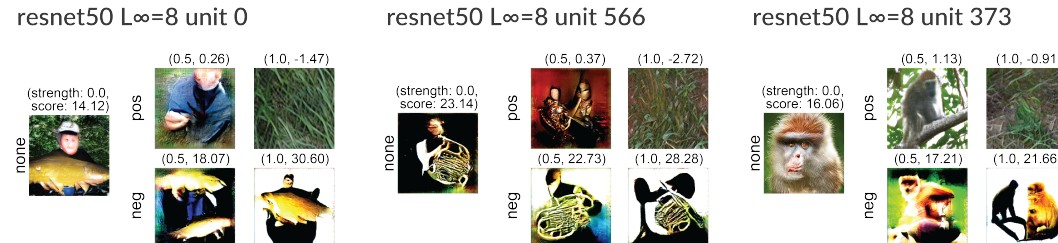

Figure 6: Robust network ResNet50 $L_\infty = 8$ shows a large change in preferred features upon input ablation

trained on ImageNet, visualizations of original features of intact units produced images containing objects, which produce a baseline "objectness" score. If ablation removes the object from the visualizations, the objectness score should decrease from baseline. Indeed, ablation of positive weights resulted in a decreased objectness score from baseline (Fig. 5). Therefore, object information is segregated to the positive weights during network training.

### 4.2 SEGREGATION DEPENDS ON RELU BUT NOT ON UNSUPERVISED PRETRAINING

To test whether training the networks under supervision for classification is required for the segregation of object information to the positive weights, we performed ablation experiments in a network trained without supervision. We used the publicly available siamese network ResNet50SimSiam (Chen & He, 2020), consisting of a ResNet50 backbone trained without supervision, then its weights frozen, it was coupled to a fully-connected layer and fine-tuned to solve ImageNet1000. This network also segregated the main features to the positive weights, but the features vanished under smaller ablation strengths than the CNNs trained on classification (Fig. 20, 19). The representation of this network also had small changes upon negative weight ablations. Thus, inputs from unsupervised representations also organize themselves such that positive weights convey most of the relevant features. We hypothesized that ReLU's rectification, which yields non-negative activations, causes a split into positive and negative weights. In ReLU networks, the weights define the sign contributions to the next layer. Maximizing a unit's activation involves enhancing positive inputs and reducing negative ones, allowing relevant features to activate positive weights and suppress negative ones. Conversely, a network using a non-rectified activation function like Tanh can encode relevant features as positive inputs with positive activations and weights, or negative activations and weights. To investigate whether functional segregation is influenced by the shape of the activation function, we trained a ResNet18 model using the Tanh activation function instead of ReLU. Unlike ReLU, the Tanh function is not rectified; it is anti-symmetric about zero and sigmoid-like, with output values ranging from -1 to 1, centered at an input value of zero. Both ResNet18 networks were trained using the FFCV library (Leclerc et al., 2023) for 16 epochs on the ImageNet 1000 dataset. The top-5 classification accuracy was 0.797 for the network with Tanh activations and 0.870 for the network with ReLU activations. The ResNet18-ReLU network behaved consistently to other networks, being more susceptible to changes upon ablation of positive weights (Fig. 21, 22). In contrast, the ResNet18-Tanh network exhibited similar changes in activity and representational similarity for both positive and negative ablations, maintaining relevant features despite the elimination of either input polarity. Thus, rectification in activation functions is a critical factor in segregating features into positive and negative weights.

### 4.3 ROBUST NETWORKS ARE LESS ROBUST TO ABLATIONS

Robust networks are better models of some aspects of biological vision. The term *robust networks* refers to networks trained to be invariant to small perturbations of its inputs, which can cause normal networks, but not humans, to misclassify the image. (Szegedy et al., 2014; Madry et al., 2019). Here, we observed the intact representations of robust networks seemed more object-like, and ablations of negative input weights resulted in background color changes, usually turning white (Fig. 6). Analyzing the effects of ablation revealed that increasing the level of robustness in training ResNet50, while increasing their robustness against adversarial attacks, also increased their vulnerability to ablations. This is seen by the larger change from control versus complete ablation of positive inputs (Fig. 7).

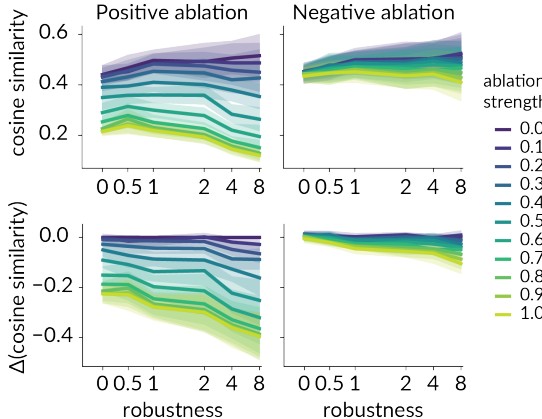

Figure 7: Representational changes upon input ablation increase with robust training for ResNet50. Top are the raw cosine similarities to control representations. Bottom are the representational changes relative to control.

Table 1: Spearman correlation of representational change upon ablation vs robustness ($L_\infty$ norm)

| type | Positive | Negative |
|---|---|---|
| $\alpha$ | $\rho$ (pvalue) | $\rho$ (pvalue) |
| 0.1 | -0.17 (2e-1) | -0.10 (4e-1) |
| 0.2 | -0.39 (1e-3) | -0.21 (8e-2) |
| 0.3 | -0.34 (4e-3) | -0.14 (3e-1) |
| 0.4 | -0.38 (1e-3) | -0.34 (5e-3) |
| 0.5 | -0.47 (6e-5) | -0.46 (9e-5) |
| 0.6 | -0.48 (3e-5) | -0.34 (5e-3) |
| 0.7 | -0.51 (9e-6) | -0.52 (6e-6) |
| 0.8 | -0.50 (2e-5) | -0.49 (2e-5) |
| 0.9 | -0.48 (4e-5) | -0.62 (2e-8) |
| 1.0 | -0.47 (6e-5) | -0.57 (5e-7) |

The robust networks also were more susceptible to ablation of negative input weights, which produce background changes, but to a lesser extent (Table 1). Overall, our results showed that classification CNNs learn to segregate object information into the positive weights and texture/background information into their negative weights, and robust training enhances this segregation.

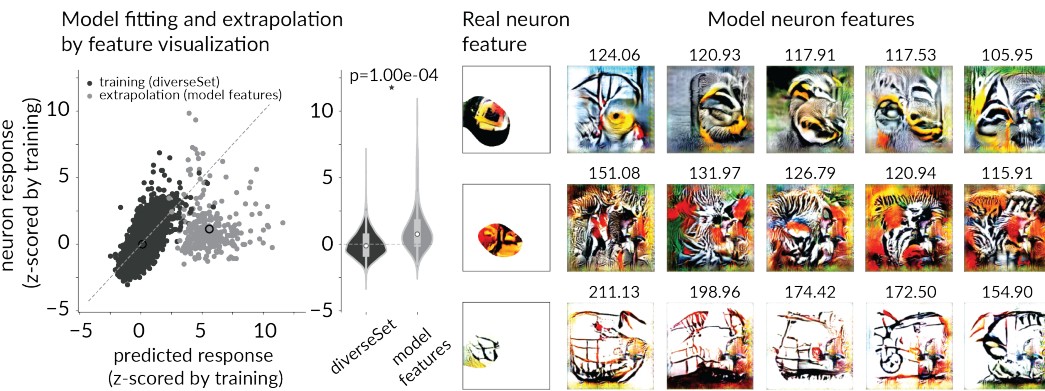

Figure 8: Neuron model units recover features relevant for the biological neurons. Left: Responses vs predicted responses of neurons to the training images, and the extrapolated features visualized from the intact models, which are extrapolations because the training data did not cover those high response ranges. Permutation t-test of neuron responses shows higher responses to images from model features than the natural images of the training dataset (diverseSet). Right: three neuron examples that show the feature visualization of the preferred feature of the neuron masked by the full-width at half-maximum obtained from perturbations to the image, and to their right the five feature visualizations of the intact model with the real neuron responses to those images on top.

### 4.4 BIOLOGICAL MODELS BASED ON IMAGENET NETWORKS SEGREGATE LOCAL FEATURE INFORMATION INTO POSITIVE WEIGHTS

Because the ventral stream in primates is thought to underlie object recognition, and recognition networks are used to model the ventral stream, we hypothesized that a similar segregation of positive and negative inputs may occur in the brain. However, our current experimental tools preclude a similar ablations as performed here in CNNs.

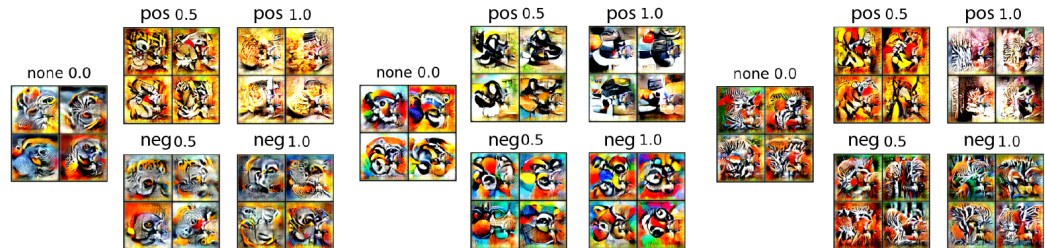

Figure 9: Preferred features of neuronal network models of visual neurons in the primate ventral stream. Pos: are positive ablations, neg are negative ablations, number indicates ablation strength.

To test this hypothesis and also whether the functional segregation in classification units extends to other tasks, we used the same inputs to classification units in AlexNet to predict responses of biological neurons in the ventral stream. We recorded responses from V1, V4, and IT cortex neurons to a set of 160 diverse images, diverseSet, uniformly covering the embedding space of AlexNet. Then, we used partial least squares to do a linear regression between each neuron responses and the activations from the penultimate layer of AlexNet (4096 units). While intermediate layers may provide better fits, we wanted to test if a simple reconfiguration of the same inputs that achieve ImageNet classification would still segregate information into positive and negative weights. We performed the ablation experiments on the best neuron models (defined by the test $r^2$ score, overall mean 0.274 with 0.096 std). Model weights had both positive and negative values with a ratio of 1.17 total positive to total negative weights (Fig. 16). The number of neurons modeled per area (V1, V4, pIT) is (7, 5, 23) for Monkey C and (1, 5, 18) for Monkey D. Thus, our results are largely representative of the pIT cortex neurons. We presented the images from the feature visualizations of the neuron models under ablations to the monkey in the same session. Because the image optimization of the model predicted neuronal responses that were larger than the responses in the training data, the models effectively performed extrapolation — we found that the optimized images of the models activated the neurons more than the training set by over one standard deviation (Fig. 8, left). When possible, we also performed feature visualization in vivo for the modeled neuron. In these cases, we found that models based on just 160 images were able to capture the preferred visual features of the neuron (Fig. 8, right). While the neuron features obtained in vivo were spatially localized (procedure in A.1), the model features obtained in silico were not not necessarily restricted to one location, and appeared in several locations, also in mirrored or rotated versions. This reflects invariances in the networks that may not exist in the neurons. Unlike the images from recognition units, images from the neuron models did not resemble objects (Fig. 8, 9). The images from the neuron models under positive input ablations elicited a consistent decrease of activity in the biological neurons providing further support to the models despite the limited training dataset (Fig. 10, left panels). Interestingly, the images from ablation experiments of single neuron models were also able to elicit changes in the average response of the neuronal population. Thus, the feature changes elicited by weight ablation translate to meaningful changes in the images perceived beyond the modeled neurons to across the ventral stream (Fig. 10, rightmost panel).

Alike the representation changes of the recognition networks, representations of the neuron models changed most with positive than negative ablations (Fig. 10). Therefore, the segregation of ablation effects are not restricted to a classification objective, but also to a regression objective, as the neuron models are just the same input activations that feed into the classification output of ImageNet CNNs reweighted via linear regression. While our dataset precluded fitting models following Dale's law, we found models using only positive weights had lower training and test performance compared to unconstrained models (Fig. 17). Thus, receiving negative inputs from the artificial network features improved the response predictions of our biological neurons. We found features that were consistently assigned positive/negative weights in most neurons models (more than 90%). Visualizing these intermediate layer features from AlexNet fc ReLU layer (4096 units) showed positive features had smaller scale edges, curvatures and spots, while negative features had more textures and larger patches (Fig. 18). These features are a hypothesis for excitatory and inhibitory neurons in IT cortex, which require genetic tools in the primate to be adressed experimentally. This suggests a functional segregation of contextual information to inhibitory inputs in high-level visual cortex, a hypothesis testable with new genetic tools in non-human primates.

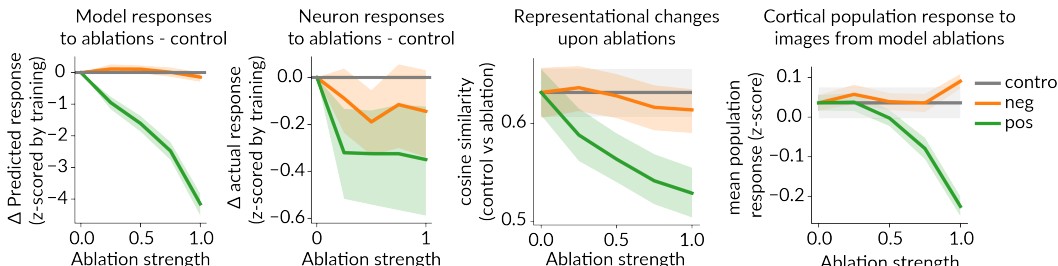

Figure 10: Left: predicted and actual neuron responses of model to ablations. Images obtained from positive ablations in the neuron models elicited a consistent activity drop on the biological neurons modeled. Right: Representational change of model to ablations measured by our cosine similarity metric on the neuron model feature visualizations upon ablation; and cortical population response to the images obtained from feature visualization from ablation of model units, neurons were z-scored before computing the population average. Plots show averages over 59 models, (35 for monkey C, and 24 for monkey D), shaded regions are the 95% C.I of the mean. For all plots the positive ablation condition was statistically different to the control.

## 5 LIMITATIONS

Our results hold in the last layer units of multiple networks. Due to limited computing time, we did not test all 1000 categories in as many networks as possible, our largest test consisted of 100 units. While larger scale simulations will provide exhaustive evidence, we are confident our main claims will stand. We limited our neuron recordings to a 160 image dataset for regressing neuron responses via CNNs. While we observed good fits and recovered relevant feature to the neurons, more images may improve the models, especially when those images are larger-scale versions of our diverseSet. The neuroscience results would need to follow Dale's law to be mapped one-to-one to excitatory and inhibitory neurons, but we make no claim to such strict mapping in this work.

## 6 DISCUSSION AND CONCLUSION

Our study combined ablations with feature visualization guided by naturalistic image priors to reveal the functional segregation of class-level features in the output layer of ImageNet trained CNNs: positive weights contribute object information, while negative weights contribute background or contextual information. This effect was enhanced in robust networks, it was present in networks with unsupervised pretraining, but was absent in network trained with Tanh instead of ReLU. Our results explain how the background contribution to classification observed in (Xiao et al., 2020) emerges, backgrounds are primarily encoded by the negative inputs.

Importantly for neuroscience, the observed functional segregation in neuron model units in CNNs hints at a functional segregation in the brain beyond the center-surround classically studied in V1. And we crafted a diverse dataset for visual neuroscience recordings that is scalable. Neuron responses to a smaller but diverse set of naturalistic, colored images, with complex foregrounds and backgrounds, led to models capturing relevant features obtained experimentally from the neuron. Thus, using both model-based and model-free approaches revealed richer neuronal representations. Preferred images from neuron models with positive input ablations elicited smaller average population responses of cortical neurons. This suggests that ablation in networks modeling neurons holds potential as a method to control the population activity in the brain. To relate ablation-induced changes in the images to the population responses is a future direction. This ablation based on the natural division of positive and negative weights can be easily extended into arbitrary layers, e.g., using gradients to define positive and negative contributions to any arbitrary unit. And our ablation approach proposes baselines for the functional differences between excitatory and inhibitory neurons in higher cortical visual areas. Understanding the circuit mechanism of biological vision could aid further understanding and development of computer vision models.

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

# A    APPENDIX

## A.1    EXTENDED METHODS

**Networks**    The ablation studies were performed on CNNs pretrained on the ImageNet dataset: AlexNet (Krizhevsky et al., 2012), VGG16 (Simonyan & Zisserman, 2015), ResNet50 (He et al., 2015), and robustly-trained ResNet50 ($L_\infty \in \{0.5, 1, 2, 4, 8\}$, Salman et al. (2020)). All these networks end on a 1000-unit fully connected layer, each unit corresponding to one of the 1000 ImageNet categories. Neural networks were used in Pytorch.

**ImageNet subsampling**    To reduce computing time, for most of the experiments, we used a subset of ImageNet, the *imagenette* dataset (Fas, 2024) and the macaque category, 11 classes in total. These classes and their corresponding output units in each network trained on the 1000-class ImageNet dataset are as follows: (0, tench), (207, English Springer), (482, cassette player), (491, chain saw), (566, church), (569, French horn), (571, garbage truck), (574, gas pump), (701, golf ball), (970, parachute), and (373, macaque). We visualized the representations of the output layer units of those classes under different ablation conditions. For Fig. 14, to sample 100 diverse classes out of the 1000 ImageNet classes, the 50k validation images were first clustered into 100 clusters via agglomerative clustering of the L2 distance matrix from the 1000-d output features of ResNet50, which was pre-trained on ImageNet. Then, one new unique class is selected from each cluster.

**Ablation**    We used two ablation conditions: we ablated weights that were (1) only positive or (2) only negative. We ablated weights cumulatively by first sorting the positive (or negative) weights by their (absolute) decreasing value. We defined the *ablation strength*, $\alpha$, as a fraction of the total positive or total negative weights to a unit. We identified the top $k$ weights necessary to reach the silencing strength, i.e., $\sum_{i=1}^{k} w_i \leq \alpha$, and set them to zero. We covered the range of ablations from 0 to 1. For most experiments with ANNs, we used silencing strengths in 0.1 steps, from 0 (intact) to 1 (complete ablation).

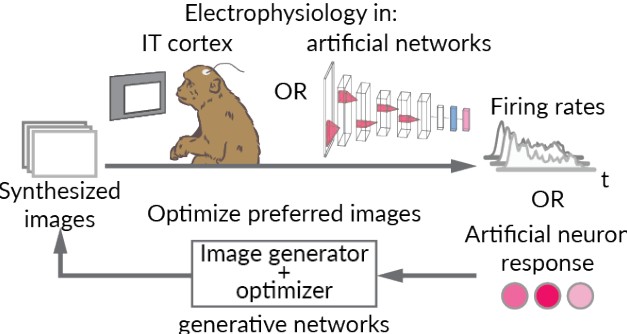

Figure 11: Schematic of feature visualization workflow in ANNs and brains. Optimizer is CMAES, image generators are DeePSim fc6 or BigGAN.

**Feature visualization**    For each ablation condition, we performed feature visualization by optimizing a GAN latent code to create an activity-maximizing image Fig. 11. We used this closed-loop, zeroth-order-search approach to allow comparison with our neuronal experiments, where gradient ascent would not be possible. To increase the span of the stimulus space, we used two GANs: AlexNet fc6 DeePSiM (Dosovitskiy & Brox, 2016) and BigGAN (Brock et al., 2019). For optimization, we used a variant of *covariance matrix adaptation evolutionary strategy* or CMAES (Wang & Ponce, 2022; Loshchilov, 2015). Initial conditions for the CMAES were given as standard deviation

## AlexNet output feature space (PCA)

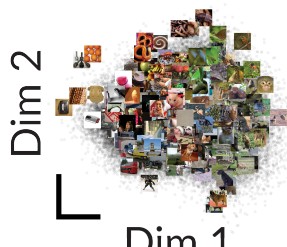

## 160DiverseSet

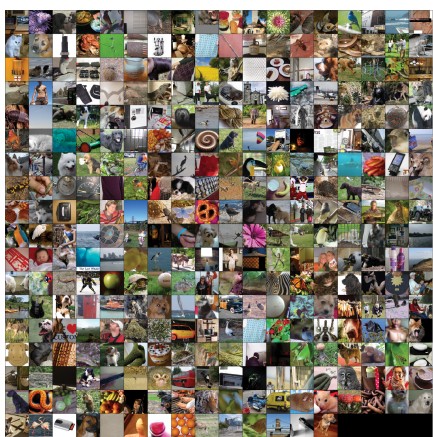

Figure 12: Illustration of a diverse dataset construction using AlexNet output feature space. The embedding is the output of the last layer before softmax of AlexNet, a vector space of 1000-dimensions. Left: PCA showing the coverage of the feature space by the diverseSet 160, only for illustration purposes. Right: images from diverseSet 160 used to fit neuron models.

of 3.0 for DeePSim, and 0.2 for BigGAN. Initial images for the algorithm were small norm vectors for both GANs, close to the origin of the latent spaces. For BigGAN, we generated a fixed noise vector by scaling a 128-dimensional truncated noise sample (-1.4, 1.4), and concatenated it with a 128-dimensional zero vector of the class embedding, to form the required 256-dimensional input code. The remaining parameters are determined by the dimensionality of the search space of each GAN. We optimized ten images per GAN, resulting in 20 feature visualizations per output unit and ablation condition. Diverse visualizations better capture the multifaceted high-level representations in CNNs (Nguyen et al., 2016b). For our examples, we show the best of the 20 visualizations, but used all for quantitative analyses. For visualizations of neural networks predicting biological neuron responses, due to experimental time restrictions, we used five visualizations per ablation condition, via DeePSim only. Our experiments are performed in a PC with Nvidia 4090 GPU, and each visualization running 100 iterations takes about 3 mins. For *in vivo* experiments, we ran from 20 to 60 iterations of the AlexNet fc6 DeePSiM with the CMAES algorithm implemented in Matlab, linked to our real-time spike-sorting data acquisition. The responses fed to the CMAES algorithm were the average firing rate on the window 70-170 ms from image onset.

**Feature analysis** We computed image similarity using an ensemble of CNNs, including AlexNet, ResNet50, and ResNet50 with robustness in $L_\infty \in \{0.5, 1, 2, 4, 8\}$, inspired by (Feather et al., 2023) And confirmed the results with LPIPS (Zhang et al., 2018) in the appendix. We computed their activations and defined similarity as the average pairwise cosine similarity (LPIPS) between control activity vs input-ablated activity. We averaged the results of the CNNs ensemble, resulting in one quantity per ablation condition. We computed *objectness* as the maximum bounding box score provided by YOLOv7 (Wang et al., 2022), this was averaged over visualizations per unit, units per network, and then across networks.

**Visual cortex electrophysiology** We collected data from two animals (monkey C and monkey D), each implanted chronically with floating multielectrode arrays (Microprobes for Life Sciences, MD) of 32 or 16 channels (monkey C, N = 96 electrodes, monkey D, 64), in areas V1, V4 and posterior inferotemporal cortex (PIT). All institutional procedures were followed. Channels were distributed as (V1, V4, PIT): monkey C (32, 32, 32), monkey D (16, 16, 32). Some electrodes captured the activity of single units, but most showed multi-unit activity (reflecting the pooled activity of micro-clusters of neurons). The animals performed a simple fixation task, which required them to keep their eyes on a 0.25-deg diameter spot at the center of the screen, within a square fixation window measuring 0.5–1° per side. Images were presented for 100 milliseconds ON, 150-ms off, 4-5 images per trial, after which the animal received water or juice. Images were presented to monkey C were

2 deg in size, and 4-8 deg for monkey D to match the receptive field centers of most channels in all cortical areas (V1, V4 and PIT). Image presentation and data acquisition (electrophysiology, eye tracking) were integrated by the MonkeyLogic2 software (Hwang et al., 2019) and OmniPlex Neural Recording Data Acquisition Systems (Plexon Inc.), interfaced through custom Matlab code. We performed online spike sorting using the PlexControl client based on waveforms. We used ViewPixx EEG monitors (ViewPixx Technologies), at a resolution of 1920x1080 pixels with 120 Hz refresh rate. Eye tracking used ISCAN cameras (ISCAN Inc.). And reward was delivered using the DARIS Control Module System (Crist Instruments).

**Feature localization in vivo**   We conducted a perturbation-based localization to identify relevant image regions from a feature visualization performed in vivo, where gradient information from the animal brain is unavailable. We perturbed a circular region with a 50-pixel diameter within the 256-pixel image by randomly shuffling the pixels inside this circle, effectively disrupting the local image structure while maintaining local contrast. We selected 30 such regions for perturbation at random, excluding those that extended beyond the image boundaries. The modified images were then presented to the monkey. We hypothesized that perturbing regions crucial for driving the neuron response would lead to a decreased firing rate. To assess local image importance, we calculated the normalized response change: the difference between the firing rate response to the intact image and the firing rate response to the perturbed image, divided by the firing rate response to the intact image. A normalized response change of 0.5 indicates the neuron response decreased by half due to perturbation. To generate the localized response mask, we averaged the circular masks corresponding to each perturbed region, weighted by their response change. This response mask was further smoothed using a Gaussian kernel with a 30-pixel standard deviation. We defined relevant regions as those causing a normalized response change of 0.5 or greater. Finally, we applied this mask to the original feature visualization image to highlight the local features.

**Image dataset**   We collected a reference image dataset to activate neurons in the monkey along the hierarchy of V1, V4, and PIT. Because neurons vary in their preferred features, we constructed a dataset spanning the image space as represented by the neural embedding of ImageNet-trained AlexNet. The embedding is the output of the last layer before softmax of AlexNet, a vector space of 1000-dimensions. The images from this dataset also spanned uniformly the 1000-dimensional output space of a semi-supervised trained network, trained on a billion images, ResNet50SS (Yalniz et al., 2019). To define this embedding space, we performed PCA on the output activations from AlexNet to the 50k ImageNet validation images, we kept the top 300 components (accounting for about 95% of total explained variance). Then we partitioned the space into a defined number of clusters $k$, according to the desired dataset size, using batched k-means to reduce computational burden. After finding the $k$ cluster centers, we could feed arbitrary images to the network, map them to the PCA space, and then pick the nearest neighbors to the cluster centers from the desired image space. In addition to the ImageNet validation set, we added other common neuroscience datasets (Brady et al., 2008; Kar et al., 2019; Allen et al., 2022; Hung et al., 2005) to form our image space. We selected $k = 160$ images, as a set that was diverse but small enough to be used in every experimental session. We called this image dataset *diverseSet* .

**Models fit on neuronal activity**   We recorded responses of neurons in the ventral stream to a 160 image dataset, our diverseSet Fig. 13. We relied on a small dataset to fit neuron responses and perform feature visualizations within the same experimental session. We performed partial least-squares linear (PLS) regression (80/20 train/test split) between the neuron responses to images and the activations of the penultimate layer of AlexNet. We used one component for the PLS regression. We selected one neuron or microcluster per experimental session, fitted a model, and performed the ablation and feature visualizations *in silico* for that model. We selected the best fitted neuron per session, based on the $r^2$ on the 20 % held out test set, usually in the range of 0.15 to 0.5. When time allowed, we also performed the feature visualization of the modeled neuron *in vivo* using a gradient-free approach (Ponce et al., 2019), within the same experimental session. To test whether features learned by the model were relevant to the biological neuron, we recorded the neuronal responses to the preferred images of the model. We then analyzed the representational similarity of the model features under ablations using ANNs. And analyzed the responses of the biological neuron populations from V1, V4 and IT.

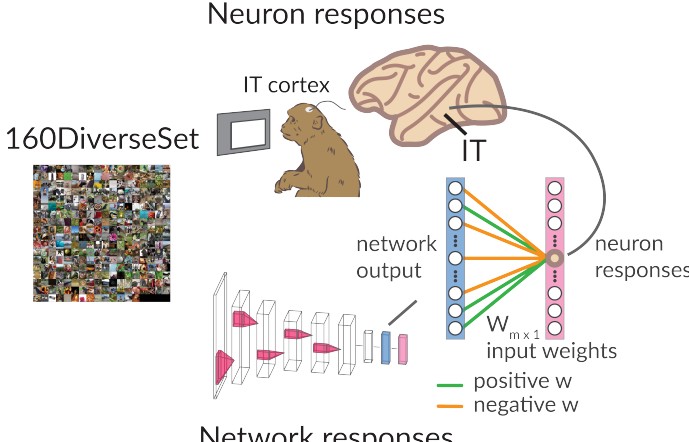

Figure 13: Schematic of model fitting using the dataset diverseSet. 160 images were split into train/test datasets (80/20).

## A.2 SUPPORTING RESULTS

Table 2: Ratio of positive to negative weights. We divided the sum of positive weights by the sum of the absolute values of the negative weights.

| Model | Ratio (mean $\pm$ std) |
|---|---|
| AlexNet | $1.03 \pm 0.08$ |
| VGG16 | $1.01 \pm 0.09$ |
| ResNet50 | $1.00 \pm 0.06$ |
| ResNet50 ($L_\infty = 0.5$) | $1.00 \pm 0.05$ |
| ResNet50 ($L_\infty = 1$) | $0.99 \pm 0.05$ |
| ResNet50 ($L_\infty = 2$) | $1.00 \pm 0.04$ |
| ResNet50 ($L_\infty = 4$) | $1.00 \pm 0.05$ |
| ResNet50 ($L_\infty = 8$) | $1.01 \pm 0.05$ |

**ResNet50 last fc layer units: 10x larger dataset, 100 new imagenet classes**

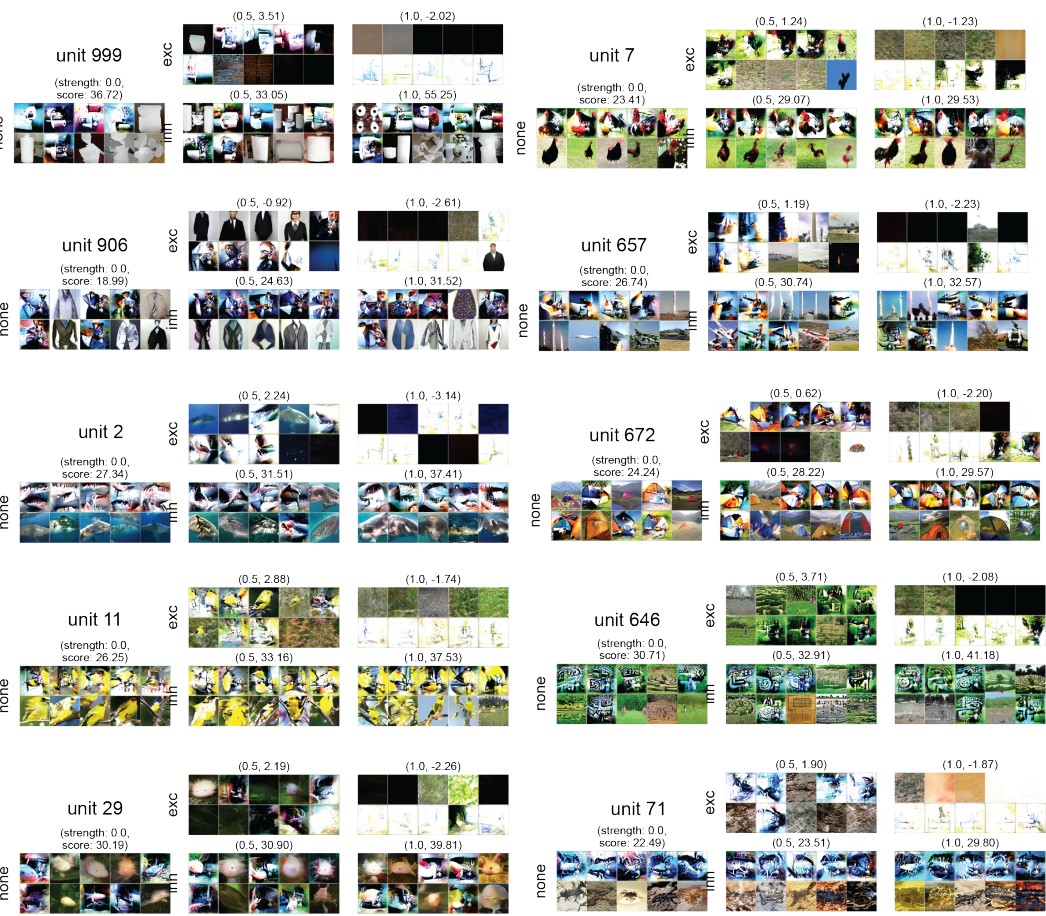

Figure 14: Functional segregation holds in a 10x larger dataset. 100 classes out of the 1000 ImageNet categories were selected by clustering the 50k validation images embedded in the 1000-d output space of ResNet50 picking one class per cluster. Thus, we now have 10x more data points that should span the representational space of the output layer we study. Consistent with the smaller dataset, the main object features degrade into more uniform background images upon positive ablation. Here we show examples from 10 of the 100 classes.

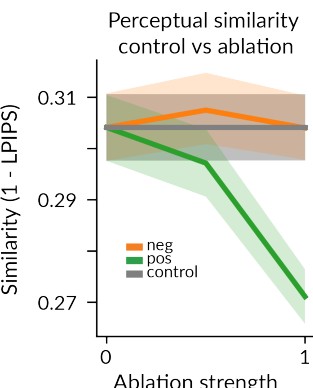

Figure 15: Functional segregation holds in a 10x larger dataset with LPIPS (Zhang et al., 2018) as representational similarity measure. We measured the representational similarity of the images as 1 - LPIPS among control images and between control images and ablation images. We average results per class, and show the mean and 95% C.I. across the 100 classes. The representational similarity degrades upon positive input ablations, confirming results obtained from the imagenette dataset.

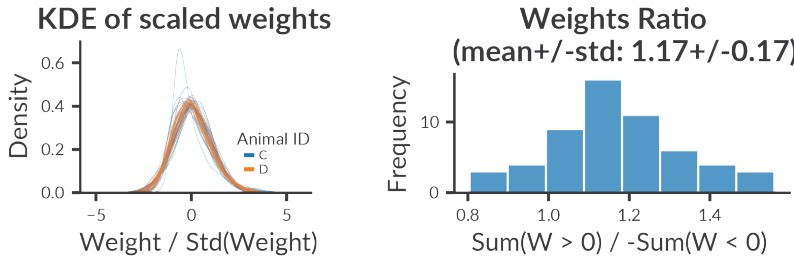

Figure 16: Left: Distribution of the model weights from neuronal fits with AlexNet penultimate layer features. Each model maps 4096 parameters from penultimate layer of AlexNet to the response of one biological neuron. Models use positive and negative weights. Model weights were normalized by their standard deviation to plot them on the same scale, for sake of visualization. Right: Ratio of total positive to total negative weights, per neuron model. Models use slightly larger positive weights with a mean of 1.17 and std of 0.17. Model numbers: 35 for monkey C, and 24 for monkey D.

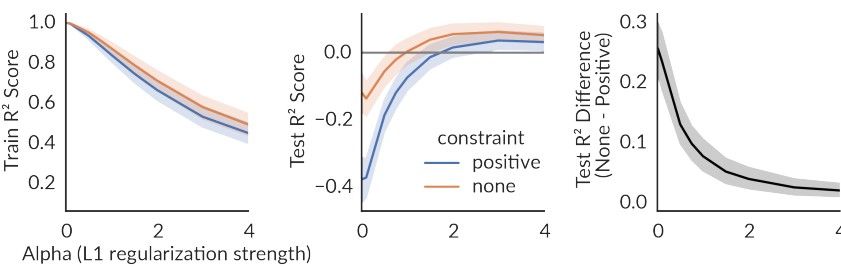

Figure 17: Using negative weights improves neuron models obtained via Lasso regression. Lasso regression models were fit with and without the positive constraint, over a 5-fold cross validation. Models were a linear regression from the 4096 features to a single neuron, over all neurons modeled from both animals. Left: performance on the training set measured by $r^2$ score. Middle: $r^2$ performance on the test set. Right: Model improvement by using positive and negative weights vs using only positive weights given by the difference in $r^2$ on the test set. Unconstrained models perform better than the positively constrained model, across the range of L1 penalties (sparseness penalty) tested, suggesting negative inputs from artificial network features are useful to predict biological neuron responses.

**Features from neuron models, AlexNet 4096 ReLU fc layer**

Positively weighted    Negatively weighted for >90% neuron models

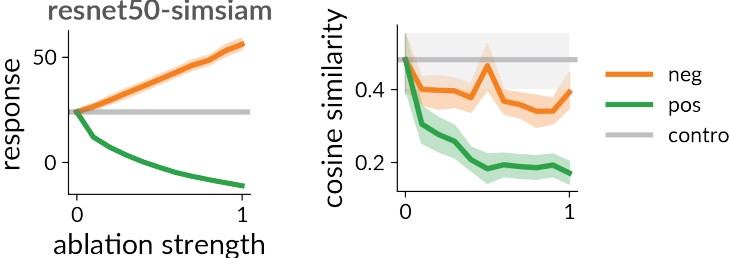

Figure 18: Features that had positive or negative weights in most of the neurons models ( 91% of the 56 neurons). These features are the closest approximation to features respecting Dale's law from our models. Left: best of 20 feature visualizations for the features with positive weights across neurons, feature index is on top of the image. Features are from the penultimate fc layer post ReLU, containing 4096 units. Right: best feature visualization from the negatively weighted features across neurons. Positively weighted features contain more local features like curved edges, while negative features contain textures or larger image patches. Sign consistency tested for statistical significance against the Bernoulli distribution of 0.5 probability with Bonferroni correction for testing 4096 features.

Figure 19: Effect of unsupervised pretraining on ablation studies using ResNet50-SimSiam. ResNet50SimSiam (Chen & He, 2020) trained without supervision, with frozen weights, was coupled to a fully connected layer, only this layer was fine-tuned to classify ImageNet1000. Left: Mean activation scores of units used in ablation experiments. Units scores come from the last fully-connected layer, with 1000 units, before the softmax. Right: Representational similarity of intact vs input-ablated units measured by the pairwise cosine similarity of control vs ablation images over an ensemble of networks. The units correspond to the 10 imagenette categories ([0, 217, 482, 491, 497, 566, 569, 571, 574, 701]) plus the macaque category (373). Error bars are 95% confidence intervals over units (categories tested), each unit response is the mean of its 10 visualizations. Control refers to the feature visualizations in the intact networks for the same units, we extended it as a horizontal line to ease visual comparisons to the different ablation strengths.

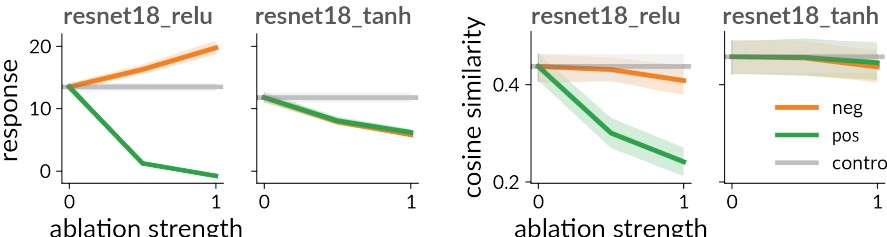

Figure 20: Feature visualizations of ablation experiments in a network pretrained with unsupervised learning. ResNet50SimSiam (Chen & He, 2020). The unsupervised network with frozen weights was coupled to a fully connected layer, only this layer was fine-tuned to classify ImageNet1000. Network units changed starting with small positive weight ablations, see unit 574 golf ball. Smaller changes are visible upon negative weight ablations, however object relevant features remain. Overall behavior is consistent with CNNs trained directly on ImageNet1000 classification.

Figure 21: Effect of nonlinearity of the activation function in ablation studies, ReLU vs Tanh in ResNet18. Left: Mean activation scores of units used in ablation experiments. For all networks, units scores come from the last fully-connected layer, with 1000 units, before the softmax. Right: Representational similarity of intact vs input-ablated units across recognition networks tested, measured by the pairwise cosine similarity of control vs ablation images over an ensemble of networks. The units correspond to the 10 imagenette categories ([0, 217, 482, 491, 497, 566, 569, 571, 574, 701]) plus the macaque category (373). Error bars are 95% confidence intervals over units (categories tested), each unit response is the mean of its 20 visualizations. Control refers to the feature visualizations in the intact networks for the same units, we extended it as a horizontal line to ease visual comparisons to the different ablation strengths.

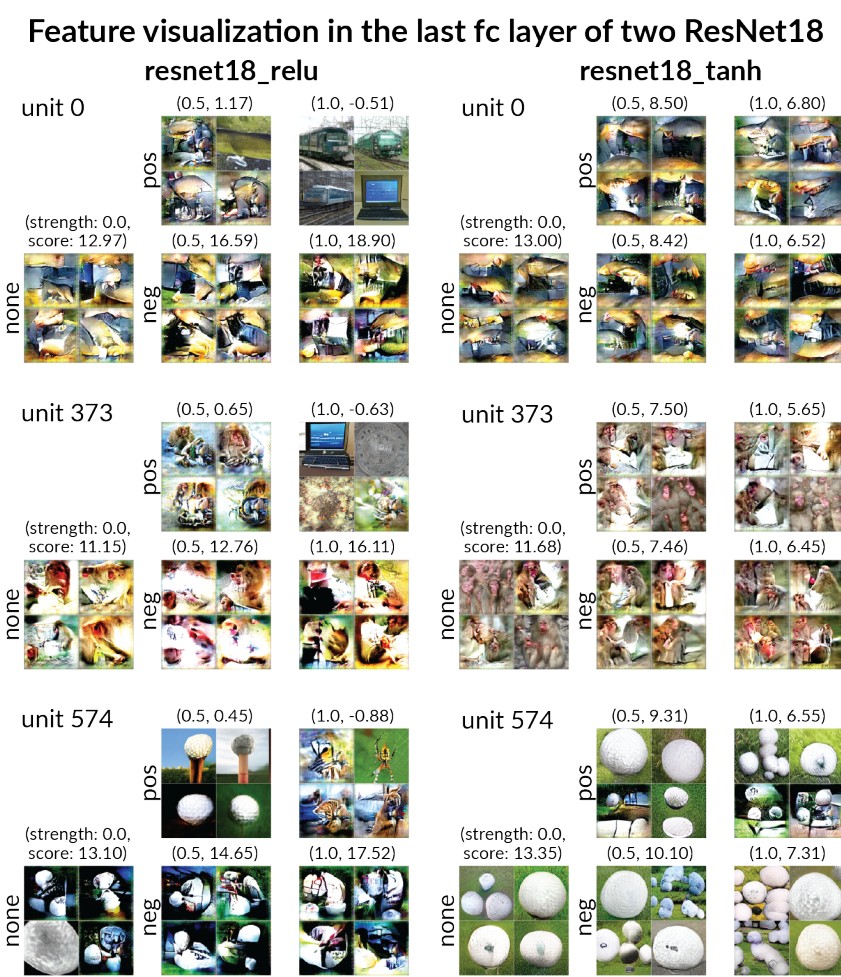

Figure 22: Feature visualizations of units in the last fc layer of ResNet18 with ReLU (left) and ResNet18 with Tanh (right) upon input ablations. ResNet18 with Tanh conserves relevant features of the corresponding categories even when all positive or all negative weights have been ablated.

