# OpenReview forum: "Functional segregation of inputs in artificial neural networks for vision"
_ICLR.cc/2025/Conference — Submitted to ICLR 2025_

### Official Review · Reviewer_YVb8 · 2024-10-28

**Soundness:** 2
**Presentation:** 2
**Contribution:** 2
**Rating:** 5
**Confidence:** 5

**Summary:**

This paper aims to find the distinct role of excitatory and inhibitory synaptic weights in biological and artificial networks. Ablating positive vs negative weights in a layer of neural network, visualizing the affected features, showed that object related features when ablating positive weights, while the background texture remains less affected. When networks mapped to real neurons in primate brain (mostly PIT area) were ablated similarly, a consistent result was reported. Altogether, this work suggests the role of inhibitory neurons is shape feature selectivity in primate vision.

**Strengths:**

Role of excitatory and inhibitory neurons in natural vision remains a fundamental question in neuroscience. On the other hand, the rise of mechanistic interpretability in ML, begs the question are they related to meaningful features in the image? The study is well-motivated, and the study of both natural and artificial visual systems side-by-side is very important to illuminate both fields.

**Weaknesses:**

#### Insufficient experiments to support the claims

> 4.1 NETWORKS TRAINED ON IMAGENET ALLOCATED OBJECT INFORMATION INTO POSITIVE
WEIGHTS

To claim, the effect seen for positive vs. negative weights in DNNs are relevant for excitatory vs inhibitory synapses in the brain, one should look into more layers and not just the layer before softmax in AlexNet ('fc' layer). That layers contain weights which during training were encouraged to organized in 'be a large positive' for 'the correct class' and be suppressed otherwise, because of the properties of the softmax. So, it is almost trivial that ablation of positive weights in that layer hurt the object features and keeps the background intact.
This explanation is perfectly inline with the next experiment that showed

> 4.2 ROBUST NETWORKS ARE LESS ROBUST TO ABLATIONS

where the results show robustness increases the segregation. Robust training makes the model to rely on object features more, as previous work showed that robust neural networks are more shape-biased (Geirhos et al, 2018). Also, see background challenge paper (Noise or Signal: The Role of Image Backgrounds in Object Recognition, Xiao et al, 2020)

So, to address this concern, I suggest running control experiments for other layers (doesn't need to be exhaustive, and doesn't need to include black-box feature visualization which is time-consuming). Just a few other intermediate layers from simple networks using simple (but reliable) gradient-basedd feature visualization would work. Inclusion of the information regarding positive/negative weight ratio is important, too.

> 4.3 BIOLOGICAL MODELS BASED ON IMAGENET NETWORKS SEGREGATE LOCAL FEATURE
INFORMATION INTO POSITIVE WEIGHTS

The main concern that I have regarding this section is that unlike the first experiment where the proportion of positive vs. negative weights were listed (Table 1), it's not clear how to interpret the results in this section without that information.

Moreover, since the main question is about the functional role of inhibitory vs excitatory synapses, here is a good chance to restrict the mapping to excitatory vs. inhibitory **neurons** as opposed to **synapses**. Because real neurons can't have both type of synapses and since the goal in this closed-loop monkey-included experiment is to uncover the role of inhibition, I wonder why not establish a Dale's law mapping network instead of regular PLS which allows positive and negative weights for all units. I appreciate that in the limitations authors brought up Dale's law, just wondering if Dale's law in mapping as opposed to Dale's law in the trained network (which is very constraining) could bring more insights about inhibition vs excitation in the brain.

In summary, the main claim in the paper about role of excitatory vs inhibitory neurons in object feature enhancement needs support because the experiment on penultimate layer where positiveness of weights are directly linked to classes can't be generalized to positive vs negative weight's role in the whole network (or brain).

**Questions:**

- In figure 9, why figures were labeled as *exc*, vs *inh* rather than *pos* vs *neg* as before? The weights are still not in a biological brain so I found this labeling a bit misleading.

---

> ### Author Response · Authors · 2024-11-30
> **Reply to first comment on DNNs**
>
> We thank the reviewer for their thoughtful feedback and the opportunity to clarify and expand on our findings.
>
> In short, our main updates are: *1) we find that ReLU is necessary for weight sign segregation; 2) unsupervised networks also segregate information similarly; 3) neuron models benefit from negative inputs; 4) we visualized DNN features that are weighted by the same sign across 90% of the cortical neurons, with positive features being local, high-contrast curvatures, spots, balls, and edges, and negative features being more diverse, e.g., textures, blobs, gradients.*
>
>
> We wanted to explore if using a brain-inspired ablation procedure could shed light on the representations of object categories in DNNs. As we found an interesting segregation, this prompted us to wonder if our work can already shed light into the visual cortex function, beyond our studies on DNNs.
>
> Training a network for classification usually involves a cross-entropy loss, deriving from softmax. Even in unsupervised learning, networks solve downstream tasks by training additional layers on top of the frozen backbone, which involves softmax for classification. This led us to explore the nature of representations that emerge in relation to signed inputs to output class units, where a unit-to-class mapping is shaped by softmax.
>
> The reviewer points out that softmax results in high activation for the correct class units, however softmax doesn't inherently dictate which features become relevant for a class. Prior studies, some cited by reviewer (Xiao et al., 2020), show there is non-trivial accuracy even when objects are removed and only backgrounds remain in the images. For instance, grassy or aquatic backgrounds are associated with specific animal or fish classes, respectively. Nonetheless, backgrounds are more variable than object features, which naturally leads to positive weights encoding object features more prominently than background information.
> While this segregation may seem trivial to the reviewer, it is unexpected that the background can also be learned by the negative features, especially in robust networks. Greater robustness tends to increase the amount of background information in negative features while making positive features more object-centric and decreasing performance. Thus, knowing how the background is encoded aids explainability of classification networks, including the effects of adversarial training. Such findings prompted us to ponder how the brain handles classification across diverse backgrounds, using features of inferotemporal (IT) neurons, which we address in our next reply.
>
>
> To investigate the causes of weight segregation, we focused on two aspects: the role of unsupervised learning and the role of activation function rectification. This makes up a new subsection in the paper with the next points:
>
> -	First, we explored whether supervised training is necessary for the segregation of object information into positive weights by conducting ablation experiments with a network trained without supervision, ResNet50SimSiam. We found that this network also segregated key features into positive weights, but these features were more sensitive to smaller ablation strengths compared to CNNs trained directly for classification (Fig. 19, 20). The network's representation showed minimal changes when negative weights were ablated, suggesting that even unsupervised inputs tend to organize such that positive weights convey most of the relevant features.
>
> -	Second, we investigated whether activation function rectification influences feature segregation, hypothesizing that ReLU's non-negative activations contribute to dividing features into positive and negative weights. In ReLU networks, maximizing a unit’s activation involves features that activate positive weights and suppress negative ones. In contrast, with a non-rectified activation function like Tanh, networks can encode relevant features as either positive or negative activations, with the matching sign weights. To test this, we trained a ResNet18 model using Tanh instead of ReLU. The ReLU network was consistently more susceptible to changes upon ablation of positive weights (Fig. 21, 22). In contrast, the Tanh network exhibited similar changes in activity and representational similarity for both positive and negative ablations, maintaining relevant features despite the total elimination of either input polarity. This suggests that information was duplicated across positive and negative weights. Rectification in activation functions is key to segregating features into positive and negative weights, influencing information distribution even within the same architecture. Classification objectives, often using softmax, drive networks to encode categories, and it is the combination of the ReLU activation and the loss function that facilitates feature segregation. Notably, ReLU's behavior mimics the neuronal threshold for action potentials.

---

> ### Author Response · Authors · 2024-11-30
> **Reply to intermediatte layers and neuron models**
>
> ### On intermediate layer features
> Prior work has shown that intermediate layers encoding less complex features, such as curvatures, receive negative weights from interpretable anti-features. For example, a curvature such as “(“ could receive inhibition from a feature “)” with opposite curvature (Olah et. Al 2020). The equivalent feature inhibition has been explored in the retina with ON-OFF interactions or in V1 with orientation selective inhibition of Gabor filters. Thus, we aimed to derive hypotheses from using object-category layers in DNNs for IT neurons, as they have not yet been studied to that level of detail. We address biological neurons further in the second reply.
>
> In our neurons models, ablation of positive weights resulted in the loss of some features, such as colors or high contrast curvatures, sometimes leaving a textured background and others leaving more defined features. Thus, we observe a commonality of visual neurons and classification networks in that positive weights are assigned to the most critical features. But we observe a slight discrepancy in that negative weights could carry different degrees of texture vs localized features for visual neurons and more background and texture features for the negative weights in classification networks. This is addressed in our next point.
>
> We performed ablation experiments in the penultimate fc layer (one before the layer we extensively studied here), not included in the pdf, where we observed that positive weight ablation resulted in a range of features from background as in the last fc layer, up to features still resembling the original but changed in color, shape or more nuanced variables. Thus, while the last layer confines negative weights to background and textures, in intermediate layers this is not the case.
>
> ### On neuron models and Dale's law
>
> We thank the reviewer to motivate a more thorough characterization of our experimental data.
>
> We included the distribution of weight ratios per model in the appendix (Fig. 16), mean is 1.17 and std 0.17, indicating a bias towards positive weights.
>
> We agree that a Dale’s law mapping would bring more insight. Because Dale’s law involves quadratic optimization and not linear one, the amount of data required surpasses our experimental capabilities. Training such a brain model is a great line of future research.
>
> We did two things to address this issue to the best of our efforts.
>
> 1.	To test the relevance of negative weights to explain neuron responses, we performed Lasso regression with and without a positivity constraint on the weights on the penultimate layer features from AlexNet (fc post ReLU 4096 units). Models with negative weights had better train and test performance than models with only positive weights for all L1 penalty values tested (Fig. 17). Thus, neural network features from the penultimate layer can function as useful negative inputs in neuron models.
>
> 2.	To better approximate Dale’s law, we analyzed the weights across the neuron models for both animals. Interestingly, we found some features that had consistent signs across more than 90% of the fitted neurons. Thus, this consistently positive/negative features could be our best hint at the properties of excitatory and inhibitory inputs to neurons in the brain. Thus, we visualized the features, which are units from the penultimate fully-connected layer of alexnet post ReLU, providing additional information about another layer in AlexNet. The consistently positive features included more localized high-contrast curvatures and edges. In contrast, the consistently negative features were more diverse, including textures and high-frequency details similar to backgrounds, and patches like blobs of uniform color. These are useful baselines and hypothesis for future neuroscience experiments, as tools to record from distinct neuron types are becoming available for primate use.
>
> ------
>
> > In figure 9, why figures were labeled as exc, vs inh rather than pos vs neg as before? The weights are still not in a biological brain so I found this labeling a bit misleading.
> - We corrected the labels to avoid misleading the reader.
>
> -------
>
> In summary, we provided insights about the relevance of ReLU vs Tanh in weight segregation for DNNs, and found that neuron models fitted in different neurons from different animals, can learn important features with consistent weight sign that can provide expected features in future neuroscience experiments.
>
> We thank the reviewer for further discussions.

---

> > ### Comment · Reviewer_YVb8 · 2024-11-30
> >
> > **Thank you for the responses.**
> >
> > I appreciate the changes to the text and the additional experiments, but they don’t directly address the main suggestion—measuring the same effects in layers other than the last layer. Instead, the authors added experiments on ReLU vs. tanh and a fine-tuned self-supervised network. These additions don’t provide much new information. The differences between the ReLU and tanh networks can mostly be explained by the lower performance of the tanh network (8% less Top-5 accuracy) and its lack of clear feature visualization. Similarly, the self-supervised experiment with a fine-tuned linear layer just confirms that the last layer assigns higher values to the correct class in a network trained for object recognition.
> >
> > The focus on studying positive and negative weights in the last layer and connecting them to excitation/inhibition in the brain is still misleading. Without further evidence, these claims are not well-supported.
> >
> > To better explore the role of excitation and inhibition in relation to their functional effects:
> >
> > **Analysis Across Layers for ANN results**: For ANNs trained on object classification (either supervised or self-supervised), show that in both internal and final layers, negative and positive weights correspond to different feature selectivities. As mentioned earlier, any simple but reliable feature visualization method (like gradient-based techniques, e.g., Adebayo et al., 2020) would work. I suggested this simple analysis in my first review.
> >
> > **Excitation vs inhibition in the IT cortex**: For recorded PIT neurons, use spiking activity to define putative inhibitory and excitatory neurons and show that excitatory neurons are selective for objects, while inhibitory neurons are selective for backgrounds.
> >
> > In its current form, the paper refines the claims, but the results on ANNs are not novel, as other reviewers have noted. The parallels to excitation/inhibition in the brain are also not backed by the provided evidence.

---

> > > ### Author Response · Authors · 2024-12-03
> > >
> > > We thank the reviewer for the reply. The feedback has led to stronger backing for our claims.
> > >
> > > ## Analysis Across Layers for ANN results
> > > To fill the gap of CNN internal layers, as suggested by the reviewer, we have now performed gradient-based feature visualization in the internal layers and found that at those layers there is a split in the features represented that are predominantly contributing positive vs negative weights to the next layer.
> > > The first layer of AlexNet is divided by positive features with achromatic high-frequency edges, negative features are lower frequency chromatic edges and spots. While for the final convolutional layer, negative features resemble backgrounds (grass, sky, people), while positive features resemble object parts (animal snouts and eyes).
> > >
> > > We have anonymously uploaded the detailed figure with caption to figshare https://figshare.com/s/2bea686a28349d598213?file=50935179
> > >
> > > While there is a difference in the features contributing positive vs negative weights, only in deeper layers features start becoming more object- and background-like, consistent with the deeper layers being closer to solving the classification task. Thus, we anticipate that excitatory vs inhibitory features in the brain could diverge along the cortical hierarchy as we observe here with depth in ANNs. In fact, our finding that Conv1 positive features are edges of high-frequency and negative ones are lower frequency edges, is consistent with V1 excitatory neurons preferring higher spatial frequencies than inhibitory neurons, as is also pointed out by other reviewer referring to modeling work (King et al. 2013). This makes us confident that this segregation has parallels in visual cortex and ANNs.
> > >
> > > Thus, we thank the reviewer again for reinforcing this suggestion that really expanded our claims beyond the final layer, making it more relatable to the brain.
> > >
> > > **Other comments about ANNs**
> > >
> > > To our knowledge, other reviewers did not raise novelty concerns. The visualizations from Tanh contain the relevant features for the classes, e.g. a fish for tench, pink faces with brown fur for macaque, and a white ball for golf ball. Thus, while there is a performance drop, the network is still able to solve the task, and does so by assigning similar features to positive and negative weights. Prior theoretical work has shown that the nonlinearity impacts learned features: "ReLU […] leads feature neurons to specialize for different regions of input space. By contrast, feature neurons in Tanh networks tend to inherit the task label structure." (Alleman, Lindsey and Fusi 2024 ICLR). This could explain why segregation depends on ReLU vs Tanh in our empirical studies.
> > >
> > > King, P. D., Zylberberg, J., & DeWeese, M. R. (2013). Inhibitory
> > > interneurons decorrelate excitatory cells to drive sparse code formation in a spiking model of V1. https://doi.org/10.1523/JNEUROSCI.4188-12.2013
> > >
> > > ## Excitation vs inhibition in the IT cortex
> > > We have defined putative excitatory and inhibitory model inputs similarly as we did for ANNs in the first part of this reply, here we find features that are positive for 90% or more of recorded neurons or negative for 90% or more neurons. Then we visualize the features, arising from the penultimate fully-connected layer of alexnet post ReLU (4096 units).
> > >
> > > - Positive features included more localized high-contrast curvatures and edges (Fig. 18, left).
> > > - In contrast, the consistently negative features were more diverse, including textures and high-frequency details similar to backgrounds, and patches like blobs of uniform color (Fig. 18, right).
> > >
> > > The positive vs negative features are more similar to the inputs to Conv5 layer than to the final layer (figure included in previous section).
> > > These are useful baselines and hypotheses for future neuroscience experiments, as tools to record from distinct neuron types are becoming available for primate use.
> > > After this additions, we expect that the differences between excitatory and inhibitory IT neuron features will have a functional segregation more similar to the later layers, but different from the fully object vs background extreme at the final classification layer.
> > > We thank the reviewer for the careful consideration of our work, and believe that we have now addressed remaining concerns.

---

### Official Review · Reviewer_PwM9 · 2024-11-04

**Soundness:** 3
**Presentation:** 3
**Contribution:** 3
**Rating:** 6
**Confidence:** 3

**Summary:**

This paper uses positive and negative weight ablation experiments as well as feature visualization to show that positive weights in a CNN tend to encode object information, while negative weights tend to encode background and contextual information, respectively.  The authors show that this is not only true for CNN but also for the neurons in the macaque ventral visual system. The authors further show that this tendency is even stronger in robust neural networks.

**Strengths:**

The study appears to be carefully organized and systematically conducted. The basic findings appear to be consistent and valid across multiple CNN models, though less so for models of biological neurons.

**Weaknesses:**

First, it is debatable whether the positive and negative weights in CNN can be equated to the excitatory and inhibitory input to a neuron or the action of excitatory neurons and inhibitory neurons. Second, in the visual cortex, such as V1, inhibitions coming from the surroundings or from within the hypercolumn in the primary visual cortex are known to mediate competition from other objects in the scene (same and different locations) as a way to resolve ambiguity. From the traditional neuroscience perspective, there is no particular reason that the inhibition has to carry only  "background" or "texture" information. Third, figure 9's ablation experiment on the neuronal model fitted to the neuron's responses did not appear to contain only background texture, even with the positive weights ablated. These concerns lead me to question whether these findings are relevant to understanding the brain.

**Questions:**

Is there a logical or computational explanation as to why the negative weights are carrying "background" and "texture" information?

---

> ### Author Response · Authors · 2024-11-30
>
> We thank the reviewer for their insightful feedback, which has significantly deepened our understanding and improved the quality of our work.
>
>
> ### Weakness:
>
> Our main findings explain a phenomenon in CNN final layers and assess the extent to which these mechanisms are used in cortical areas responsible for object classification.
>
> We agree with our Reviewer that it is debatable whether the positive and negative weights in a CNN can be equated to excitatory and inhibitory inputs to a neuron. It would be unwise to claim that, which is why the term “analogous” is used to draw parallels. However, there is a dearth of hypotheses about the roles of inhibitory and excitatory neurons when it comes to visual tuning and processing. The best contribution of CNN-based studies to neuroscience is to generate hypotheses that can be tested in vivo. Here, our computational results elevate the hypothesis that object-related critical features are more likely to be associated with long-range projections in the primate visual system (where only excitatory neurons project across areas, e.g. V4  to IT), whereas local, intra-areal inhibitory neurons (e.g. IT-IT) might combine long-range projections to represent correlated contextual information. Specifically, this correlated information might be a background texture in an object-background classification layer, such as fc8, *or* other localized critical features ancillary to the primary objective, as we find on the models fitted to cortico-visual activity. So, while we generally agree with our Reviewer that there is no particular reason why inhibitory neurons must only carry a certain type of information, no one has thoroughly demonstrated what kinds of information can or should be carried by inhibitory neurons. Motivated by our findings (new below), our next in vivo project will apply the same (gradient-free) method and modelling in V1, V4, and PIT neurons in combination with novel methods to record from excitatory or inhibitory neurons in primates.
>
> In reply to the comment, we analyzed of our neuron models to establish a closer analogy to excitatory and inhibitory contributions. By examining the weights across neuron models in both animals, we identified features with consistent signs in over 90% of fitted neurons. We visualized these features, representing units from the penultimate fully-connected layer of AlexNet post-ReLU, offering additional information about another layer in AlexNet.
> The consistently positive features included more localized high-contrast curvatures and edges (Fig. 18). In contrast, the consistently negative features were more diverse, including textures and high-frequency details similar to backgrounds, and patches like blobs of uniform color. This suggests that, unlike neural network units representing object categories, the brain does not exclusively associate negative inputs with background information. These findings provide valuable baselines and hypotheses for future neuroscience research.
>
>
> ### Question:
>
> While we hope our work now provides more insights into neuroscience, we have also contributed to a better understanding of convolutional neural networks (CNNs). We added a new section 4.2, demonstrating that the segregation into positive and negative weights depends on the activation function rather than on unsupervised versus supervised learning.
>
> -	Unsupervised learning using a ResNet50 backbone, followed by fine-tuning on ImageNet classification, led to similar weight segregation as observed in a vanilla ResNet50.
> -	Training a ResNet18 with tanh instead of ReLU resulted in object information being represented in both positive and negative weights.
>
> A plausible explanation is that classification demands high activations for the correct category in the final layer. When ReLU is used, relevant features must activate positive weights and suppress negative weights (ideally to zero). In principle, object features could suppress negative weights, but this requires learning complex features near zero, the ReLU threshold, below which the gradient is zero, complicating optimization. Optimization is easier when above zero, which could explain why positive weights can learn object features in their positive activations, while negative weights primarily learn background and textures. Backgrounds can be informative of some broader classes of categories (sky vs grass vs water), and vary within categories (dogs indoors vs outdoors), providing more training examples to learn from, which may explain their representation in negative weights.
> In contrast, when tanh is used, high-class activation can be achieved through both high positive activations of positive weights and high negative activations of negative weights. Since tanh is anti-symmetric, optimization is similar for both positive and negative activations. Consequently, the network can distribute relevant features across both positive and negative weights.
>
>
> We look forward to the Reviewer’s reply.

---

> > ### Comment · Reviewer_PwM9 · 2024-12-02
> > **Thank you for the responses**
> >
> > Thank you for taking the time to respond carefully. I will suspend my disbelief and increase my score by 1 to 6.  The explanations you offered about "why" positive weights carry more object information are plausible.
> > However, you should know that
> > 1. There are V2 and V4 neurons that are tuned to textures, and most likely, they are excitatory
> > (see e.g. https://www.jneurosci.org/content/jneuro/39/24/4760.full.pdf,
> > https://www.nature.com/articles/s41467-024-50821-z )
> > 2. Incidentally, the way you described the foreground and background info reminds me of a quantitative measure called "dispersity" used in  https://www.nature.com/articles/s41467-024-50821-z .
> > 3. There is literature indicating that inhibitory neurons may receive preferential input from "fast visual pathways (Chen et al. 2007), and even direct input from LGN or top-won inputs from the prefrontal cortex representing an initial guess (Bar 2003) -- that is,  information carried by inhibitory neurons is of  "lower resolution" but not necessarily "background information".  It is thought that the coarse information carried by the inhibitory neurons has shorter latency (Mruczek and Sheinberg 2012) and can influence the fine-detailed visual processing by the excitatory neurons.
> > 4. In King et al. (https://www.jneurosci.org/content/jneuro/33/13/5475.full.pdf)'s V1 model for learning sparse codes, the excitatory neurons learned Gabor, but the inhibitory neurons also learned Gabor filters though of lower spatial frequency -- consistent with point 3 above. While you can argue that the low spatial frequency is consistent with "background information", there could be many other reasons for that as well. For example, there are fewer inhibitory neurons (1/4 of the excitatory) in V1, thus the inhibitory neurons have to integrate and average a lot of information in order to fulfill its redundancy reduction role.
> > That is not to say your hypothesis is not plausible, but there might be other reasonable hypotheses as well.
> > Hope it helps.

---

> > > ### Author Response · Authors · 2024-12-03
> > >
> > > We thank the reviewer for the consideration of our reply. The references are very useful discussion points.
> > >
> > > 1. Our finding in artificial networks draws parallels between textures and backgrounds due to the object classification objective, but we agree that dispersed features or textures are also encoded by excitatory neurons in V2 and V4. Because the brain solves more tasks beyond classification it could have more flexible representations than classification networks.
> > >
> > > 2. This measure of dispersity will prove useful for feature attribution experimentally, and is related to our occlusion approach in the appendix. As we understand, those experiments recorded responses of both excitatory and inhibitory V4 neurons, thus it provides further plausibility to our hypothesis that inhibitory neurons in V4 would be more "disperse" than excitatory neurons.
> > >
> > > 3. Very interesting points. As inhibitory neurons come in different subtypes, we expect some will fulfill diverse roles, including the ones mentioned by the reviewer. For example somatostatin interneurons are associated with top-down modulation, while VIP interneurons modulate between inhibitory and excitatory neurons.
> > >
> > > 4. While CNNs break Dale's law, we found features that projected with higher sign consistency to the next layer, and visualized the top positive and negative projecting features https://figshare.com/s/2bea686a28349d598213?file=50935179 . Interestingly, AlexNet Conv1 layer recapitulates the findings referred by the reviewer (King et al.) supporting our positive vs negative feature segregation also in intermediate layers. Namely, positive features are mostly achromatic high-frequency edges, while negative features are chromatic lower frequency edges and spotted textures. Thus, in CNNs, as layers get deeper and features become more complex, the segregation that arises from low vs high frequency in the first layer turns into object vs background in the last layer. Thus, divergence of excitatory vs inhibitory neurons could increase along the ventral hierarchy.
> > >
> > >
> > > We are again grateful for the feedback, we hope the reviewer can find the discussion as enriching as we did. While there are many biologically plausible interpretations, we find that feature segregation (not restricted to just object vs foreground) is common to both brains and machines. Overall, the reviewer comments enriched the biological plausibility of the findings for other layers than IT.

---

### Official Review · Reviewer_6fyK · 2024-11-04

**Soundness:** 3
**Presentation:** 2
**Contribution:** 3
**Rating:** 6
**Confidence:** 4

**Summary:**

This paper employs sophisticated methods to investigate the roles of positive and negative weights in deep neural networks. Through feature visualization, the authors illustrate the effects of ablating positive and negative weights, finding that positive weights contribute significantly to object representation, whereas negative weights primarily encode background information. Furthermore, similar results are obtained with real neuron responses as objective, showing a potential similar mechanism also in biological visual processing.

**Strengths:**

•  The idea is novel and intriguing, addressing a fundamental problem in systems neuroscience.
•  The experiments are thorough and well-designed.
•  The results are clearly presented.

**Weaknesses:**

The paper requires a relatively advanced understanding from readers; the writing could be improved with more intuitive explanations.

**Questions:**

1.	My main concern is the rationale behind the roles of positive and negative weights. Do we have a theory explaining why positive weights contribute to object representation and negative weights to background information? Could it not be the other way around? I realize this is challenging to answer, but a direction for future research would be helpful.

Writing-related questions:

2.	In Figure 1, are the lines averaged across 10 units from the 1,000 categories? Also, what is "control" here? My understanding is that it represents no ablation, so is it expected to be a flat line?

3.	The section on ablation has some issues. Could you clarify $\sum_{i=1}^{k} w_i$? It seems wrong as alpha is the proportion. Should it be $\frac{\sum_{i=1}^{k}}{\sum w_i}  $
4.	In the Figure 3 caption, what does “visualization score” mean?

5.	In Section 4.2, what is meant by “robust networks”? Could you clarify what they are robust to?

6.	The phrase “the diverseSet covers the embedding space of AlexNet” is unclear. What does "embedding" refer to here, and which layer do the embeddings belong to?
7.	Figure 8 is confusing. What does extrapolation mean in this context? The main text does not seem to cover this—did I miss something?
8.	I understand that the neuro features obtained in vivo were spatially localized. How were these localized features obtained?

---

> ### Author Response · Authors · 2024-11-26
>
> We thank the reviewer for the careful feedback. We clarified the text for better readability and performed additional experiments to enhance insights into the work.
>
> Reply 1:  To investigate why positive weights contribute to object representations rather than the background, we trained networks with different non-linearities: ReLU and Tanh. In ReLU networks, activations at each stage are positive or zero, forcing the sign of weights in subsequent layers to determine contributions. For classification, ReLU networks must assign positive weights to class-relevant features while suppressing detrimental features through negative weights. Conversely, networks with non-rectified activations like Tanh, which is anti-symmetric and outputs values between -1 and 1, can use both positive and negative weights to amplify or suppress features. For instance, positive weights amplify features with positive activations, while negative weights amplify features with negative activations.
>
> To test this, we trained ResNet18 models using Tanh and ReLU activations on ImageNet 1000 for 16 epochs (top-5 accuracy: 0.797 for Tanh, 0.870 for ReLU). ReLU networks showed the expected susceptibility to positive weight ablations. However, Tanh networks showed symmetric effects for ablations of both positive and negative weights, with object-like features assigned to both polarities. The Tanh networks maintained representational similarity and robustness after eliminating either polarity. These results suggest that rectification in activation functions was a critical factor in segregating features into positive and negative weights. We find this to be a key insight for the function of these networks, and this brings an interesting constraint towards a more biologically plausible model of the visual system. This results are included in a new section in the main text, and Fig. 20, 21 in the Appendix.
>
> Reply 2: We corrected the legend to solve the problem, adding "Error bars are 95% confidence intervals over units (categories tested), where each unit response is the mean of its 20 visualizations. Control refers to the feature visualizations in the intact networks for the same units, we extended it as a horizontal line to ease visual comparisons to the different ablation strengths."
>
> Reply 3: We corrected the mistake pointed out in our ablation equation.
>
> Reply 4: Apologies, corrected "visualization score" to "activation score"
>
> Reply 5: We edited the text:
> "Robust networks are better models of some aspects of biological vision. The term “robust networks” refers to networks trained to be invariant to small perturbations of its inputs, which can cause normal networks, but not humans, to misclassify the image.""
>
> Reply 6:  We edited the text to:
> "The embedding is the output of the last layer before softmax of AlexNet, a vector space of 1000-dimensions. The images from this dataset also spanned uniformly the 1000-dimensional output space of a semi-supervised trained network, trained on a billion images, ResNet50SS"
>
> Reply 7: We apologize for the missing information.
> The revision now reads: "Because the image optimization of the model predicted neuronal responses that were larger than the responses in the training data, we are effectively performing successful extrapolation. The optimized images of the models activated the neurons more than the training set by over one standard deviation (Fig. 8, left)."
>
> Fig. 8 legend now reads: "Responses vs predicted responses of neurons to the training images, and the extrapolated features visualized from the intact models, which are extrapolations because the training data did not cover those high response ranges."
>
> Reply 8: We added a section on the appendix:
> “Feature localization in vivo:We conducted a perturbation-based localization to identify relevant image regions from a feature visualization performed in vivo, where gradient information from the animal brain is unavailable. We perturbed a circular region with a 50-pixel diameter within the 256-pixel image by randomly shuffling the pixels inside this circle, effectively disrupting the local image structure while maintaining local contrast …”
>
> We appreciate the insights prompted by these questions.

---

### Author Response · Authors · 2024-12-04
**Common reply and acknowledgment to reviewers**

Dear Reviewers and Area Chair,

We would like to express our sincere gratitude for your valuable feedback and thorough reviews of our work.

Our findings reveal that in classification networks, the output layer differentiates objects from backgrounds through respective positive and negative input weights. We show this segregation occurs in ReLU but not Tanh networks. Furthermore, positive vs negative feature segregation begins with low-level features in the first layer and becomes distinctly object versus background only in layers closer to the output. We provide evidence, from *in vivo* recordings in primates and neuron models, supporting a hypothesis that primate vision may employ excitatory and inhibitory neurons in functionally distinct ways, similar, though not identical, to the segregation of positive and negative features in ANN layers.

The insights you provided have been instrumental in refining our study and strengthening our contributions. We are encouraged by the recognition of our work as ‘novel and intriguing,’ addressing a fundamental problem in systems neuroscience (Reviewer 6fyK), and by its systematic and carefully organized approach (Reviewer PwM9). Additionally, the acknowledgment of our study as well-motivated and important for illuminating both natural and artificial visual systems (Reviewer YVb8) has reinforced our commitment to bridging these fields.

### Summary of Revisions and Outcomes:
- **Potential mechanisms for segregation:**
We've enhanced the explanations around the role of positive and negative weights in neural networks, particularly in relation to object and background information in the penultimate layer. This includes new experiments showing how ReLU and Tanh activation functions influence weight segregation, new section 4.2 and Fig. 21, 22.


- **Examining feature segregation in intermediate layers:**
Additional experiments, as suggested by reviewer YVb8, were conducted on intermediate layers of neural networks, which confirmed feature segregation. Using gradient-based feature visualization, we showed that features that mainly have positive or negative weights to next layer also segregate in internal layers. The first convolutional layer of AlexNet is divided by positive features with achromatic high-frequency edges and negative features with lower frequency chromatic edges and spots. In deeper layers, e.g., in the final convolutional layer, negative features resemble backgrounds (grass, sky, people), while positive features resemble object parts (animal snouts and eyes). https://figshare.com/s/2bea686a28349d598213?file=50935179 .
Thus positive vs negative feature segregation begins with low-level features and becomes distinctly object versus background only in layers closer to the output.

- **Approximating Dale's law in ANN regression models of visual cortical neurons:**
We have defined putative excitatory and inhibitory model features as features that are positive or negative for more than 90% of the neurons modeled. Positive features included more localized high-contrast curvatures and edges (Fig. 18, left). Consistently negative features were more diverse, including textures and high-frequency details similar to backgrounds, and patches like blobs of uniform color (Fig. 18, right). This suggests that, unlike neural network units representing object categories, the brain does not exclusively associate negative inputs with background information. Thus, the degree of feature segregation in IT neurons may be less pronounced compared to that observed in the output units of classification networks.


The manuscript was restructured for clarity, with enhanced figures and appendices detailing empirical findings. We believe these revisions strengthen our contribution to the intersection of artificial intelligence and neuroscience, reinforcing our core claims.

We deeply appreciate your feedback, which has been crucial in shaping this refined version. Thank you for your constructive evaluation, and we warmly welcome any further thoughts you might have.

---

### Meta-Review · Area_Chair_Y8yy · 2024-12-21

**Metareview:**

This paper uses positive and negative weight ablation experiments as well as feature visualization to show that positive weights in a CNN tend to encode object information, while negative weights tend to encode background and contextual information, respectively.
When networks mapped to real neurons in primate brain (mostly PIT area) were ablated similarly, a consistent result was reported.
Taken together, this work suggests the role of inhibitory neurons is shape feature selectivity in primate vision, and offers inspiration for future in-vivo studies.

Strengths: Role of excitatory and inhibitory neurons in natural vision remains a fundamental question in neuroscience. The study is well-motivated, and the study of both natural and artificial visual systems side-by-side is very important to illuminate both fields. The experiments are thorough and well-designed. The results are clearly presented, and the findings are interesting.

Weaknesses: Reviewers have expressed doubts about the presumed connection between positive and negative weights in CNNs and excitatory and inhibitory neurons in the brain. To evaluate the contribution, this paper requires a relatively advanced understanding of neuroscience, which might make it less suitable for the ICLR audience.

This paper is outside my area of expertise and difficult for me to evaluate, and reviews place it as a borderline paper. The paper clearly has merit, and presents possibly quite relevant results for the study of the primate visual cortex. However, given the focus of the paper, I lean slightly towards rejection, and would encourage a resubmission to another venue with a stronger focus on neuroscience such as NeurIPS.

**Additional Comments On Reviewer Discussion:**

The discussion mostly revolved around the implications and interpretations of the the presented experiments, and if they are sufficient to warrant conclusions about the role of inhibitory and excitatory neurons in the primate visual cortex. The authors have provided additional results to address concerns, and agree that the connection is speculative. They emphasize (rightly so) that the role of these neurons is an open problem in neuroscience, and that their work should be seen as using artificial neural networks to provide testable hypothesis for the study of the brain. Most reviewers seem carefully optimistic about the paper.

---

### Decision · Program_Chairs · 2025-01-22

Reject